# Weibull S-N Fatigue Strength Curve Analysis for A572 Gr. 50 Steel, Based on the True Stress—True Strain Approach

**Alejandro Molina \*, Manuel R. Piña-Monarrez and Jesús M. Barraza-Contreras** 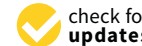

Industrial and Manufacturing Department of the Engineering and Technological Institute,
Universidad Autónoma de Ciudad Juárez, Cd. Juárez, Chihuahua 32310, Mexico;
Manuel.pina@uacj.mx (M.R.P.-M.); al187061@alumnos.uacj.mx (J.M.B.-C.)

\* Correspondence: al187118@alumnos.uacj.mx

**Abstract:** In this paper a Weibull methodology to determine the probabilistic percentiles for the S-N curve of the A572 Gr. 50 steel is formulated. The given Weibull/S-N formulation is based on the true stress and true strain values, which are both determined from the stress-strain analysis. For the analysis, the Weibull $\beta$ and $\eta$ parameters are both determined directly from the maximum and minimum addressed stresses values. The S-N curve parameters are determined for $10^3$ and $10^6$ cycles. In the application, published experimental data for the CSA G40.21 Gr. 350W steel is used to derive the true stress and true strain parameters of the A572 Gr. 50 steel. Additionally, the application of the S-N curve, its probabilistic percentiles and the Weibull parameters that represent these percentiles are all determined step by step. Since the proposed method is flexible, then it can be applied to determine the probabilistic percentiles of any other material.

**Keywords:** reliability; stress-strain analysis; strength analysis; fatigue life reliability analysis; Weibull distribution

## 1. Introduction

For the A572 Gr. 50 steel material, a probabilistic S-N curve does not exist. The main goal of the S-N curves analysis is related to the strength material behavior and its correlation with the defined stress ratio [1–4]. Structural elements are subjected to a range of stress values, so to determine the probabilistic S-N curve, any material steel is necessary [1–3,5,6]. Despite this, currently, in structural design, two different approaches have been used, and they are: (1) the application of a failure theory to determine if the designed element is safe or not; and (2) the fatigue approach [1,7]. Unfortunately, these approaches are not very efficient for predicting the reliability of a structural element. This mainly because, while the failure theory does not consider the time in the analysis, as it is the case of the Goodman, Elliptical and Soderberg failure theories given by Budynas and Nisbett [8], in the fatigue strength analysis, the S-N curve only represents the strength average [9,10]. Therefore, neither the failure theory nor the fatigue analysis is efficient to perform a probabilistic analysis [2,11,12]. Thus, in this paper based on the A572 Gr. 50 steel true stress-true strain analysis, the method to determine the probabilistic percentiles of the S-N curve is formulated, based on the two parameter Weibull distribution. Therefore, the efficiency of the proposed method to determine the probabilistic percentiles is based on the fact that the Weibull shape $\beta$ and scale $\eta$ parameters are both directly determined from the maximum and minimum material strength $(\sigma_1, \sigma_2)$ values, which are given from the true stress-true strain analysis [13]. Moreover, the analysis is based on the true stress-true strain approach, because the failure of a structure does not depend only on the applied axial stress, but it also depends

on the instantaneous area ($A_i$) and on the instantaneous elongation ($L_i$) of the element [14]. Thus, to consider $A_i$ and $L_i$ in the analysis, it was performed based on the true stress-true strain analysis.

Accordingly, due to the proposed probabilistic method, the Weibull $\beta$ and $\eta$ parameters are both determined directly from the maximum $\sigma_1$ and minimum $\sigma_2$ strength values, then based on the relationships between ($\beta$ and $\eta$) with ($\sigma_1$ and $\sigma_2$), and the corresponding log-mean ($\mu_x$) and the log-standard deviation ($\sigma_x$) values were both also estimated. Then, these $\mu_x$ and $\sigma_x$ values were used to determine the probabilistic S-N percentiles of the A572 Gr. 50 steel material. Fortunately, since the Weibull $\beta$ and $\eta$ parameters are directly fitted from the maximum and minimum strength ($\sigma_1$ and $\sigma_2$) values, then $\beta$ and $\eta$ always represent them, implying the derived $\mu_x$ and $\sigma_x$ values are both unique [13]. Therefore, by applying the proposed methodology, $\mu_x$ and $\sigma_x$ are both unique, so they can be used to determine the corresponding S-N percentile for any other material where $\sigma_1$ and $\sigma_2$ are known. However, it is important to note that, if the proposed method is going to be used to determine the probabilistic percentiles for another material, we must first be sure the used $\sigma_1$ and $\sigma_2$ stresses values are those values that generate the failure in the material, or equivalent if the used data is a failure testing data [4,6]. On the other hand, it is important to highlight that, according to the Canadian standard CSA G40.21, the A572 grade 50 steel is somewhat equivalent to the CSA G40.21 Gr. 350W steel. Thus, based on the excellent experimentation performed by Arasaratnam, Sivakumaran and Tait [15], in this paper the probabilistic A572 Gr. 50 steel S-N curve is determined by using the true stress-true strain experimental data determined from the CSA G40.21 Gr. 350W steel data, published in the True Stress-True Strain Models for Structural Steel Elements paper. The main finding of the proposed method was that the S-N construction for any material is possible, and that because the true stress-true strain analysis defines the strength limits to be analyzed, then the fitted $\beta$ and $\eta$ parameters completely represents them.

The structure of this paper is as follows: in Section 2, the fatigue strength behavior of the A572 Gr. 50 steel is given. Section 3 presents the Weibull/S-N curve formulation, based on the fatigue strength analysis for A572 Gr. 50 steel. In Section 4, the proposed method to determine the probabilistic S-N fatigue strength curve directly from the Weibull distribution analysis is given. In Section 5, the fatigue strength S-N curve application case is performed. Finally, in Section 6, the conclusions are given.

## 2. Fatigue Strength Behavior of the A572 Gr. 50 Steel

Since, by its own nature (e.g., the fabrication process and lack of homogeneity in the material) the material strength is random [16], its behavior must be modelled by using a probability density (*pdf*) function. On the other hand, as it is known, the steel materials are classified as metal and alloys, in which the A572 Gr. 50 steel is into the iron and steel group [14]. Additionally, this material is classified as a high strength-low alloy, and it is named as a typical ASTM steel [17,18]. Moreover, if the fatigue strength analysis can be used to determine the lifetime for any ductile material, and because the applied cyclic loading produces a fatigue damage process, then the probabilistic S-N curves is necessary. Due to this, it enables us to predict the cycle to failure in a probabilistic way [5,11,14]. Fortunately, from Piña-Monarrez (2017), we know that any stress-based analysis can efficiently be analyzed by using the Weibull distribution. Therefore, because the reliability of any structural element depends on the applied stress, and on its inherent strength to overcome the applied stress, then, in structural design, knowing the material strength is critical [14]. Furthermore, because, in based strength design, the used material is selected based on experimental tests data [19], represented by its corresponding S-N curve, and/or based on the applicable norm, then the material's properties considered in the analysis are the ultimate tensile strength $S_{ut}$, the yield point of materials $S_y$, the reduced tensile strength $S'_e$ and the fatigue slope of the S-N curve. Hence, based on these material properties in the next section, we present the stress-strain analysis to determine the A572 Gr. 50 steel strength behavior.

### 2.1. True Stress-True Strain Analysis for the A572 Gr. 50 Steel

Due to the material's strength throughout the differential length $dx$ of the element does not depends on the elongated area only, but it also depends on the elongated length, then the true stress-true strain analysis allows us to take into account these two variables [13,14,16]. Therefore, because for ductile materials, as it is the case of the A572 Gr. 50 steel material, by using a stress-strain analysis, the upper strength limit occurs when the material begins to be plastically deformed, then its initial cross section area $A_o$ is lowered, while its length $l_0$ is increased. In order to consider the $A_o$ and $l_0$ variables in the analysis, it must be performed based on the true stress-true strain approach. This is possible by considering that the total stressed volume remains unchanged as $A_0 l_0 = A_i l_i$, where the $A_0$ and $l_0$ are the initial cross section area and the initial length respectively. Therefore, the differential strains under the loaded component, can be estimated as the reason of the differential displacements and the initial length $l_0$ as:

$$d\epsilon = \frac{dl}{l} \tag{1}$$

Hence, all the strains behavior throughout the elongation is given by:

$$\epsilon_t = \int_{l_0}^{l} \frac{dl}{l} = \ln\left(\frac{l}{l_0}\right) = \ln\left(1 + \epsilon\right) \tag{2}$$

where the strain $\epsilon_t$ is called true strain and $\epsilon$ represents the engineering strain obtained directly of the experimental test. Based on the Arasaratnam et al. (2011) analysis, the true stress-true strain relation can be analyzed as:

$$\sigma'_t = \sigma(1 + \epsilon_t) \tag{3}$$

where $\sigma'_t$ and $\epsilon_t$ are the true stress and true strain values related to the experimental tested $\sigma$ value. Now, based on the above analysis, let present the fatigue strength analysis based on which the formulation of the S-N curve of the A572 Gr. 50 material is given.

### 2.2. S-N Fatigue Strength Analysis for A572 Gr. 50 Steel

Since the objective is to determine the maximum $\sigma_1$ and minimum $\sigma_2$ material strength values of the A572 Gr. 50 steel to perform the corresponding probabilistic analysis, then the fatigue strain-life method approach is used to define the nature of the fatigue behavior [1,2,6,8,20]. Due to the fatigue strength depends on the applied stress and on its inherent strength to overcome the applied stress, then the nature of the strength behavior throughout the lifetime structural component is analyzed by using the total strain amplitude [8,14,16,20]. Therefore, based on the Budynas and Nisbett (2008) method, the specific fatigue strength value at a specific number of cycles is estimated as a $\left(S'_f\right)_N = \frac{E\epsilon_e}{2}$, where $E$ corresponds to the elasticity modulus, $\epsilon_e$ represents the elastic strain behaviour of the material in the strain-life methodology [14,15]. Therefore, the fatigue strength at any $N$ cycles is given by:

$$\left(S'_f\right)_N = \sigma'_t(2N)^b \tag{4}$$

where $N$ represents the corresponding cycle to failure, and $b$ represents the corresponding fatigue slope given by:

$$b = -\frac{\log\left(\frac{\sigma'_t}{S_e}\right)}{\log\left(2N_e\right)} \tag{5}$$

Additionally, based on the Budynas and Nisbett (2008) and Lee et al. (2005) methodologies, because $\left(S'_f\right)_{10^3} = fS_{ut}$, then as shown in Figure 1 the $f$ value that represents the $S_{ut}$ proportion at $N = 10^3$ cycles is given as:

$$f = \frac{\sigma'_t}{S_{ut}}\left(2 \cdot 10^3\right)^b \tag{6}$$

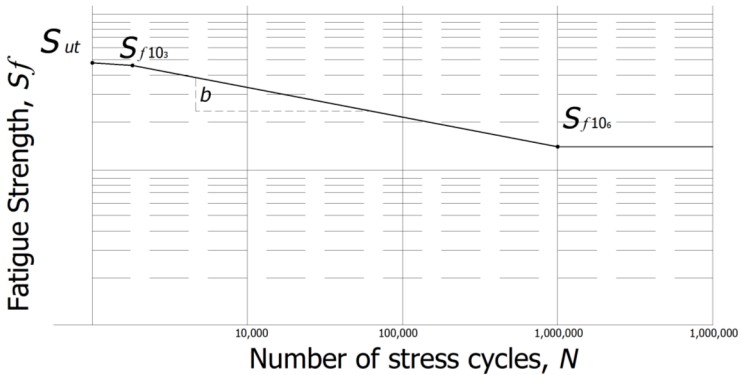

**Figure 1.** S-N diagram plotted from the fatigue strength analysis.

Following the above S-N fatigue strength analysis, the proposed method to design the A572 Gr. 50 S-N curve is now presented.

### 2.3. Proposed Method to Determine the A572 Gr. 50 S-N Curve

This proposed method to determine the S-N curve is based on the experimental data given by the Arasaratnam et al. (2011) research, where the CSA G40.21 Gr. 350W steel and the A572 Gr. 50 steel were considered to present the same properties. The steps are as follows.

*Step 1.* Determine the $S_y$, $S'_e$, $S_e$ and $S_{ut}$ values and the strain hardening exponent $n$ value of the A572 Gr. 50 steel. These values can be found in the engineering handbook [8] Appendix A, Table A-22, [9] Chapter 12, or based on the Arasaratnam et al. (2011) methodology.

*Step 2.* Take the material strain $\epsilon_{ut}$ value as the exponent $n$ value determined in step 1.

***Note 1:*** Remember that $\epsilon_{ut} = n$ holds only on the $S_{ut}$ coordinate. For any other coordinate, the corresponding $\epsilon_i$ value must be estimated.

*Step 3.* By using the material strain $\epsilon_{ut}$ value of step 2 in Equation (2), determine the true strain $\epsilon_t$ value that corresponds to the $S_{ut}$ value.

*Step 4.* By using the true strain $\epsilon_t$ value of step 3 in Equation (3), determine the corresponding true stress $\sigma_t$ value.

*Step 5.* By using the $\sigma'_t$, $S_e$ values and $N = 10^6$ cycles in Equation (5), determine the fatigue slope $b$ value.

*Step 6.* By using the $\sigma'_t$, $b$ values and $N = 10^3$ cycles in Equation (4), determine the fatigue strength $(S'_f)_{10^3}$ value.

*Step 7.* By using the above data, draw the S-N curve.

Now to determine the probabilistic S-N curve percentiles for the A572 Gr. 50 steel, let us first, based on the maximum $(S'_f)_{10^3}$ and on the minimum $(S'_f)_{10^6}$ values, determine the corresponding Weibull shape $\beta$ and scale $\eta$ parameters on which we determine the corresponding S-N percentiles. The analysis is as follows.

## 3. Weibull/S-N Fatigue Strength Analysis for A572 Gr. 50 Steel

The Weibull parameters $W(\beta, \eta)$ used to determine the corresponding probabilistic S-N percentiles are estimated as follows.

### 3.1. Fatigue Strength Analysis Based on Weibull Approach

Based on Piña-Monarrez [21] and on the addressed maximum $(S'_f)_{10^3} = S_{max}$ and minimum $(S'_f)_{10^6} = S_{min}$ values, the Weibull scale parameter is directly given by:

$$\eta_S = \sqrt{S_{max}\, S_{min}} \tag{7}$$

*Note 2:* from Equation (7), as mentioned by Piña-Monarrez (2019), the Weibull scale parameter does not depend on unknown variables, therefore it completely represents the $(S'_f)_{10^3}$ and $(S'_f)_{10^6}$ values.

Here, based on the above analysis, the log strength mean $\mu_{gS}$ value is given by:

$$\mu_{gS} = \ln(\eta_S) = \frac{(\ln(S_{max}) + \ln(S_{min}))}{2} \tag{8}$$

On the other hand, based on Piña-Monarrez (2017), to estimate the Weibull shape parameter $\beta$, first we need to determine the desired reliability $R(t_n)$ for the analysis. Then, to determine the related sample size $n$ value as:

$$n = -\frac{1}{\ln(R(t_n))} \tag{9}$$

*Note 3:* Notice that, in Equation (9), $R(t_n)$ represents the reliability of the analysis. It does not represent the reliability of the designed material.

Then, by using the $n$ value given in Equation (9), the $y_i$ elements of the $Y$ vector are determined based on the median rank approach as:

$$y_i = \ln(-\ln(1 - (\frac{n_i - 0.3}{n + 0.4}))) \tag{10}$$

Therefore, based on the aforementioned analysis, the Weibull $\beta$ parameter is determined as:

$$\beta_W = \frac{-4\mu_Y}{0.9762 \ln(\frac{S_{max}}{S_{min}})} \tag{11}$$

where $\mu_Y$ is the mean of the $Y$ vector defined in Equation (10). Thus, the $\eta_S$ value determined in Equation (7), and the $\beta_W$ parameter from Equation (11) are the Weibull parameters that represents the maximum and minimum strength values. However, to incorporate in the Weibull analysis, the effect that the behavior of the $Y$ vector has on the $\eta_S$ parameter, it is determined as:

$$\eta_W = \exp\left\{\ln(\eta_S) - \frac{\mu_Y}{\beta_W}\right\} = \exp\left\{\mu_{gS} - \frac{\mu_Y}{\beta_W}\right\} \tag{12}$$

where $\mu_Y$ represents the corresponding arithmetic mean of the $Y$ vector defined in Equation (10). Therefore, the corresponding predicted failure times can be determined as:

$$t_{Wi} = \exp\left\{\frac{y_i}{\beta} + \ln(\eta_W)\right\} \tag{13}$$

Finally, it is important to highlight that, from section 3.3 of Piña-Monarrez and Ortiz-Yanez [22], the log standard deviation of the $t_{Wi}$ elements defined in Equation (13) is directly given by using the standard deviation $\sigma_y$ the value of the $Y$ vector defined in Equation (10). Thus, by using the estimated $\beta_W$ value, the corresponding $\sigma_g$ is given as:

$$\beta_W = \frac{\sigma_Y}{\sigma_g} \quad \therefore \quad \sigma_g = \frac{\sigma_Y}{\beta_W} \tag{14}$$

Now, based on the above Weibull analysis, the fatigue/Weibull method to determine the Weibull $\beta$ and $\eta$ parameters that are used to determine the probabilistic S-N curve is as follows.

*3.2. Proposed Fatigue Strength Based Weibull Method*

Following step 7 of Section 2.3, and based on the above Weibull formulation, the steps to determine the Weibull parameters is as follows.

*Step 8*. Based on the determined $S_{max}$ and $S_{min}$ values, which represent the $(S'_f)_{10^3}$ and $(S'_f)_{10^6}$ values, respectively, from Equation (7), determine the strength Weibull eta $\eta_S$ value. As shown in Equation (8), by taking the logarithm of $\eta_S$ determine the corresponding strength log mean $\mu_{gS}$ value.

*Step 9*. Define the desired $R(t_n)$ index to be used, then, from Equation (9), determine the corresponding sample $n$ value.

*Step 10*. By using the $n$ value in Equation (10), determine the corresponding $y_i$ elements. Then determine their mean $\mu_Y$ and their standard deviation $\sigma_Y$ values.

*Step 11*. Based on step 10, determine the corresponding Weibull parameters $W(\beta_W, \eta_W)$ by using Equations (11) and (12).

*Step 12*. Based on the $\sigma_y$ value determined in step 10, and on the estimated $\beta_W$ value, from Equation (14), determine the corresponding log standard deviation $\sigma_g$ value.

Now, based on the log standard deviation $\sigma_g$ value of step 12, let us determine the percentiles for the S-N curve.

## 4. Probabilistic S-N Fatigue Strength Curve

In this section, the generalities of the Weibull and S-N curve data to formulate the probabilistic S-N curve for the A572 Gr. 50 steel are given.

### 4.1. Generalities of S-N Percentiles Formulation

The formulation to build a probabilistic S-N curve is based on the Weibull reliability function which in linear form is given as:

$$y_i = \beta[\ln(\eta) + \ln(N_i)] \tag{15}$$

Therefore, the predicted cycle to failure times $N_i$ to determine the percentiles of the S-N curve are given as:

$$\ln(N_i) = \frac{y_i}{\beta} + \ln(\eta) \tag{16}$$

From the $n$ predicted $\ln(N_i)$ elements determine their standard deviation $\sigma_g$ value. Notice this $\sigma_g$ value is the same determined in Equation (14). Thus, based on the $\sigma_g$ value and on the $d(Z)$ value, which is the value of the normal distribution that corresponds to the desired confidence level (for example to $CL = 0.95$, $d(Z) = 1.6448536$), the percentile for the $(S'_f)_N$ value is given as:

$$d(\sigma_g) = \exp\left\{\ln\left[(S'_f)_N\right] \pm d(Z)\sigma_g\right\} \tag{17}$$

Based on Equation (17), the method to determine the probabilistic percentiles for the S-N curve of the A572 Gr. 50 steel is as follows.

### 4.2. Proposed Method to Determine the Probabilistic S-N Fatigue Curve

Following step 12 of Section 2.3, the steps to determine the probabilistic S-N curve for the A572 Gr. 50 steel are:

*Step 13*. Determine the desired percentile and determine the corresponding $d(Z)$ value.

*Step 14*. By using the strength $(S'_f)_N$ value, the $d(Z)$ value and the $\sigma_g$ value in Equation (17), determine the corresponding percentile $d(\sigma_g)$ values of the S-N curve.

*Step 15*. Draw in the S-N plot the addressed $d(\sigma_g)$ values and highlight their corresponding percentiles.

Finally, notice the $\sigma_g$ value to determine the S-N percentiles determined from Equation (14) and observe this $\sigma_g$ value is unique, due to the Weibull $\beta$ parameter is also unique. Therefore, to accurately determine the Weibull parameters is critical in this analysis. Fortunately, since $\beta$ (see Equation (11)) and $\eta$ (see Equation (7)) only depends on the maximum and minimum applied stress values, then, clearly,

the accurate estimation of the maximum and the minimum stresses values is the real key variable in this analysis. The application case is now presented.

## 5. Fatigue Strength S-N Curve Application Case

The objective of this section is performing the S-N curve for the A572 Gr. 50 steel and its corresponding probabilistic S-N curve analysis, based on the Weibull distribution. Therefore, the step by step numerical application is as follows. All strength data is given in *MPa*.

*Step 1.* The selected material's features for A572 Gr. 50 [15,17,18] are $S_y = 344.73$, $S'_f = 225.11$, $S_{ut} = 450.22$ and $S_e = 131.72$.

*Step 2.* Based on the experimental data given by Arasaratnam et al. (2011), the corresponding value of the strain hardening exponent parameter is $n = 0.1511$, which represents $\epsilon_{ut}$ in the $S_{ut}$ cordinate of the strain curve.

*Step 3.* By using Equation (2), the corresponding true strain at the $S_{ut}$ coordinate is $\epsilon_t = 0.1407$.

*Step 4.* Based on the estimated true strain, the corresponding true stress $\sigma_t = 513.58$.

After this true stress-true strain analysis, the numerical data for the standardized S-N curve is given below.

*Step 5.* Based in the strength analysis given above, by using Equation (5), the fatigue slope value is $b = -0.1115$.

*Step 6.* Based on the determined true stress $\sigma_t$ value of step 4, the corresponding proportion of the *f* value from Equation (6) is $f = 0.00337$.

*Step 7.* Similarly, by using the determined true stress $\sigma_t$ value in Equation (4), the corresponding strength for N = $10^3$ and $10^6$ are $(S'_f)_{10^3} = 220.10$ MPa and $(S'_f)_{10^6} = S_e = 131.72$ MPa.

Here, by using the corresponding values of N = $10^3$ and $10^6$ the standardized S-N diagram is given in the Figure 2.

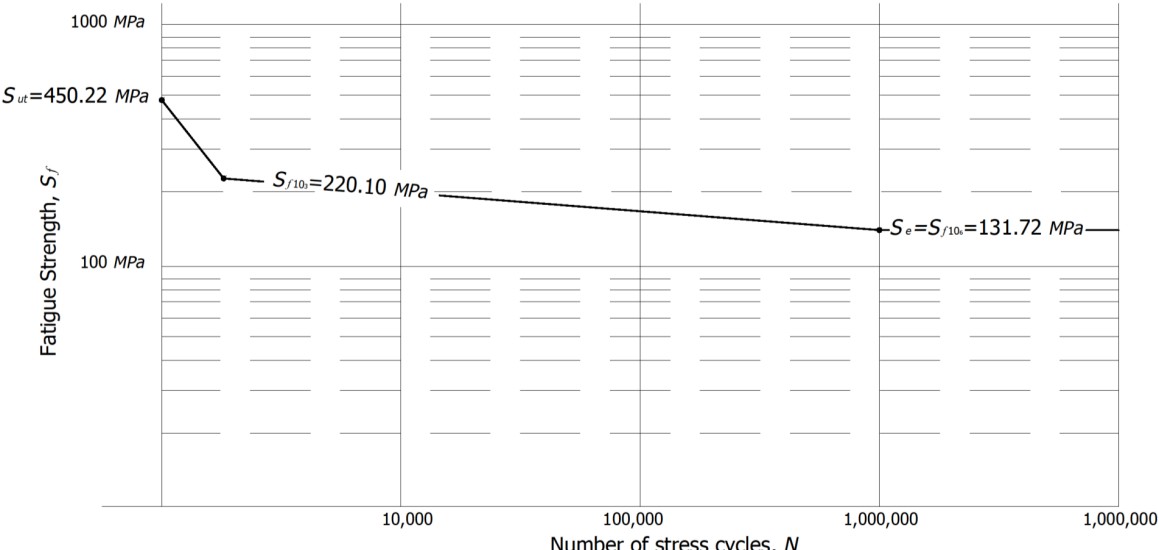

**Figure 2.** S-N diagram for application case.

Now, the probabilistic percentiles for the Weibull S-N diagram are given below. All strength data is given in *MPa*, and the log values are dimensional values based on the estimated *MPa* data.

*Step 8.* By using the estimated values in step 5 and 6, and based on the fact that the $(S'_f)_{10^3}$ and $(S'_f)_{10^6}$ values represent the $S_{max}$ and $S_{min}$ values respectively, by using Equation (7) the strength eta value is $\eta_S = 170.27$, and from Equation (8) the log strength mean value is $\mu_{gS} = 5.13$.

*Step 9.* The desired reliability for the analysis is $R(t) = 95.35\%$. Thus, from Equation (9), $n = 21$. The generated *Y* elements, the corresponding mean and standard deviation are given in Table 1.

**Table 1.** Mean $\mu_Y$ and standard deviation $\sigma_Y$.

| STEP 10. EQ. 9: $N$ | 1 | 2 | 3 | 4 | 5 | 6 | 7 |
|---|---|---|---|---|---|---|---|
| STEP 10. EQ. 10: $y_i$ | −3.4034833 | −2.4916620 | −2.0034632 | −1.6616459 | −1.3943983 | −1.1720537 | −0.9793812 |
| STEP 10. EQ. 9: $N$ | 8 | 9 | 10 | 11 | 12 | 13 | 14 |
| STEP 10. EQ. 10: $y_i$ | −0.8074473 | −0.6504921 | −0.5045088 | −0.3665129 | −0.2341223 | −0.1052851 | 0.0219284 |
| STEP 10. EQ. 9: $N$ | 15 | 16 | 17 | 18 | 19 | 20 | 21 |
| STEP 10. EQ. 10: $y_i$ | 0.1495258 | 0.2798450 | 0.4159621 | 0.5625020 | 0.7276158 | 0.9293107 | 1.2296598 |
| STEP 10. $\sum y_i - \mu_Y$ | −0.5456241 | | | | | | |
| STEP 10. $\sum y_i - \sigma_Y$ | 1.17511694 | | | | | | |

*Step 11.* Based on the estimated $\mu_Y$ and $\sigma_Y$ values given in the Table 1, from Equation (11) and 12, the estimated Weibull parameters are $\beta_W$ = 4.354995038 and $\eta_W$ = 193.00 MPa, respectively.

*Step 12.* Based on step 11 and by using Equation (14), the estimated log standard deviation value is $\sigma_g$ = 0.269832.

Based on the estimated Weibull $(\beta_W, \eta_W)$ parameters, the corresponding 95% and 5% percentiles are:

*Step 13.* For this application case, the selected percentiles to the $(S'_f)_{10^3}$ values were the 95% and 5% percentiles.

*Step 14.* By using Equation (17), the 95% and 5% percentiles values are:

$$\left[ (S'_f)_{10^3} \rightarrow d = 95\% \rightarrow d(\sigma_g) = 343.06 \right] - \left[ (S'_f)_{10^3} \rightarrow d = 5\% \rightarrow d(\sigma_g) = 141.21 \right]$$

Similarly, the corresponding 85% and 15% percentiles are:

*Step 13.1* As a second application case, the values the 85% and 15% percentiles were selected.

*Step 14.1* From Equation (17), the 85% and 15% percentiles values are:

$$\left[ (S'_f)_{10^3} \rightarrow d = 85\% \rightarrow d(\sigma_g) = 291.12 \right] - \left[ (S'_f)_{10^3} \rightarrow d = 15\% \rightarrow d(\sigma_g) = 166.40 \right]$$

*Step 15.* As a result of this analysis, the obtained Figure 3 show the percentiles analysis for the fatigue strength $(S'_f)_{10^3}$ cycles.

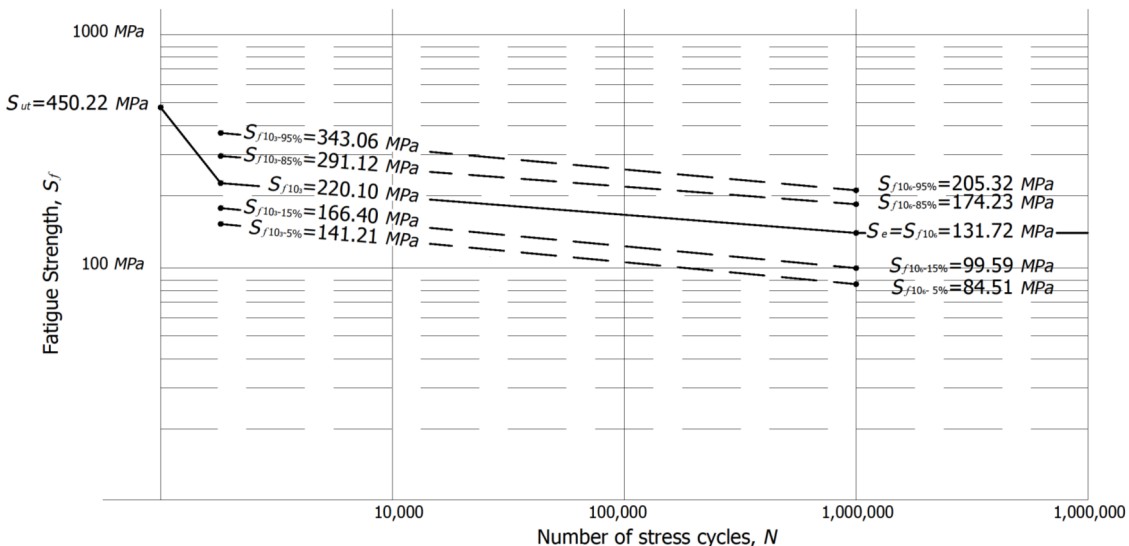

**Figure 3.** Probabilistic S-N curve for the A572 Gr. 50 steel, based on the Weibull distribution.

***Note 4:*** By following this proposed method, let us present the analysis of the fatigue strength $(S'_f)_{10^6}$ cycles. The estimated data is given (all strength data is given in *MPa)*:

$$\left[(S'_f)_{10^6} \rightarrow d = 95\% \rightarrow d(\sigma_g) = 205.32\right] - \left[(S'_f)_{10^6} \rightarrow d = 5\% \rightarrow d(\sigma_g) = 84.51\right]$$

$$\left[(S'_f)_{10^6} \rightarrow d = 85\% \rightarrow d(\sigma_g) = 174.23\right] - \left[(S'_f)_{10^6} \rightarrow d = 15\% \rightarrow d(\sigma_g) = 99.59\right]$$

As summary of this section we have that in this research, based on the true stress-true strain analysis the strength limits of the A572 Gr. 50 steel were estimated by using the conventional fatigue strength methodology. Then, by using these strength limits was possible to perform the corresponding Weibull analysis, based on which the probabilistic percentiles of the S-N curve were determined.

Now, since the reliability of an element depends on the applied stress and on its inherent strength to overcome the applied stress, let us present the corresponding stress-strength analysis.

### 5.1. Stress-Strength Analysis for a Given Structural Component Based on the Probabilistic S-N curve

For the stress-strength analysis application case, the selection's analysis of the W beam subjected to a uniform load is performed by using the available flexural strength verification method [20]. The selection case is supposed to be a left end fixed-right end free and continuously braced case. The variables for this analysis are the clear length $L = 9.00$ m, and the uniform load $W = 50.72$ KN m. After the selection process of the W beam, the selected structural component was the *W30X391*.

Then, in order to perform the corresponding stress-strength analysis, from the static analysis and based on Kececioglu [23] methodology, the alternating stress was taken to be $S_a = \sigma_x$, where $\sigma_x$ represents the estimated normal stress, and the mean range stress value was taken to be $S_m = \sigma_\mu$. Consequently, by considering the alternating stress $S_a$ is the failure mode, and that it follows a normal distribution with arithmetic mean $\mu_{s_a} = S_a = \sigma_x$ and standard deviation is given by:

$$\sigma_{S_a} = 0.10\,\mu_s \tag{18}$$

The estimated maximum and minimum expected values are:

$$\text{Maximum stress limit} = \mu_{s_a} + (\sigma_{s_a}) \tag{19}$$

$$\text{Minimum stress limit} = \mu_{s_a} - (\sigma_{s_a}) \tag{20}$$

Now, by using the above analysis based on the selected bending beam determination, the alternating stress values is $\sigma_x = \mu_{s_a} = 83.09$ MPa. Thus, the normal stress distribution used in the stress-strength analysis is $N_s(83.09,\ 8.30)$. From Equations (19) and (20), the maximum and minimum alternating stresses are $S_{a-max} = 91.40$ MPa and $S_{a-min} = 74.78$ MPa. Hence, by using these maximum and minimum values in Equation (7), the Weibull scale parameter is $\eta_s = 82.68$ MPa. Similarly, by using $\mu_Y = -0.545624$, and the maximum and minimum values in Equation (11), the Weibull shape parameter is $\beta_W = 11.14116988$. Therefore, the Weibull stress distribution used in the stress-strength analysis that represents the alternating stress behavior is $W_s(11.14116988,\ 82.6829)$.

From Section 5, the Weibull strength distribution is $W_S(4.354995038,\ 170.27)$, which is found by using the reliability function for the Weibull/Weibull stress-strength analysis Piña-Monarrez [13] given by:

$$R(t) = \frac{\eta_S^\beta}{\eta_S^\beta + \eta_s^\beta} \tag{21}$$

where $\eta_S$ and $\beta$ are the Weibull strength parameters, and $\eta_s$ is the Weibull stress eta parameter. Then, by considering the failure mode by fatigue, and by using $\beta = 4.354995038$, $\eta_S = 170.27$ MPa and $\eta_s = 82.6829$ MPa, the estimated reliability of the structural component is $R(t) = 95.87\%$.

To summarize this application case, the selected structural component was selected by using the nominal flexural strength analysis. Then, the alternating stress was estimated, and the determination of the stress rank enabled us to determine the stress Weibull parameters, which where were used to perform the stress-strength analysis. Notice that, because the probabilistic analysis indicates that the structural component is subjected to the failure, then, in the next section, the failure theory analysis is given.

### 5.2. Failure Theory Analysis by Using Probabilistic S-N curve

Additionally, the confidence level of 95% with lower limit of the strength in the $N = 10^6$ cycles is estimated below. This analysis is based on the fact that the Goodman failure theory [8] is delimited by the fatigue strength limit value $S_e$. Thus, by using the probabilistic S-N percentiles, the analysis can be determined by performing this proposed method. The analysis is as follows:

*Step 13.2* The selected percentiles for this application case are 95% and 5% for $(S'_f)_{10^6}$.
*Step 14.2* By using Equation (17), the corresponding percentiles values are

$$\left[ (S'_f)_{10^6} \ \rightarrow \ d = 95\% \ \rightarrow \ d(\sigma_g) = 205.32 \right] - \left[ (S'_f)_{10^6} \ \rightarrow \ d = 5\% \ \rightarrow \ d(\sigma_g) = 84.51 \right]$$

Now, by using this confidence level of 95% with lower limit value as the fatigue strength limit, the Goodman failure theory can be used to determine if the structural component is whether safe or not. The analysis is shown in Figure 4.

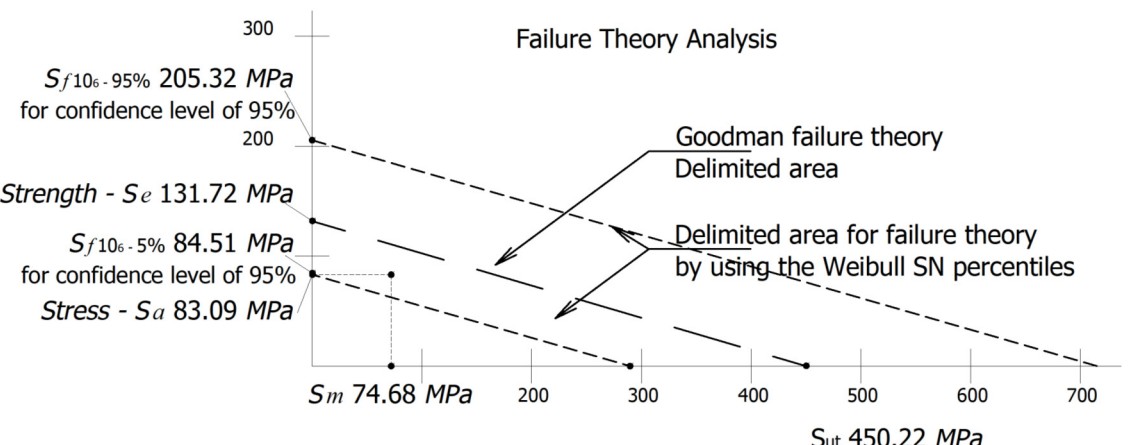

**Figure 4.** Goodman failure theory analysis.

As a summary of this section, we have that the determined lower strength limit of $(S'_f)_{10^6}$ can be used to define the safe region of the failure theory. Therefore, as a future research, it seems to be possible to set the 95% and 5% percentiles as the percentiles of the probability distribution function that model the random behavior of the fatigue lifetime strength $(S'_f)_{10^6}$ value, which can be used to model the vibration spectrum of the stresses given by the application of the structural component, but more research must be undertaken.

### 6. Conclusions

1. The proposed methodology let us to formulate a S-N curve and to determine its desired probabilistic percentiles, as well as to determine the designed reliability, as it is made for the A572 Gr.50 steel analyzed in this paper.

2. In the proposed methodology, steps 1–7 let us to determine the maximum and minimum strength S-N values, based on which Weibull parameters are determined. Steps 8–12 let us to determine the β and η parameters, as well as the log-standard deviation used to determine the S-N percentiles. Steps 13–15 let us to determine the desired S-N percentiles.

3. The stress/strength analysis given in Section 5.1 let us determine the reliability of the designed component. The formulation of Section 5.2 lets us perform the failure theory analysis by using the derived lower S-N percentile as the lower allowed value in the analysis.

4. Since the proposed methodology let us analyze variant stress, then it will be useful not only for designers, but also for structural and mechanical practitioners.

5. The log standard deviation $\sigma_g$ value used to formulate the probabilistic S-N curve percentiles is determined from the minimum and maximum stresses values, so its value is unique.

6. Although the focus in this paper was to determine the probabilistic S-N curve for the ASTM A572 Gr.50 steel material, it can be used to determine the probabilistic S-N curve for any material.

7. The key features of the given S-N methodology are: (a) the derived S-N curve is based on the true stress-true strain analysis; (b) the Weibull distribution analysis is based on the maximum and minimum strength values given by the fatigue strength methodology; and (c) the Probabilistic S-N curve percentiles given by the corresponding log-mean ($\mu_g$) and log-standard deviation ($\sigma_g$) values are based on the $\lambda_1$ and $\lambda_2$ values.

**Author Contributions:** Conceptualization, A.M., M.R.P.-M., J.M.B.-C.; methodology, A.M., M.R.P.-M.; data analysis, A.M., J.M.B.-C.; writing—original draft preparation, A.M., M.R.P.-M.; writing—review and editing, A.M., M.R.P.-M., J.M.B.-C.; supervision, M.R.P.-M.; funding acquisition, A.M., M.R.P.-M. All authors have read and agreed to the published version of the manuscript.

**Funding:** This research received no external funding.

**Conflicts of Interest:** The authors declare no conflict of interest.

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
