# Peer review of "Weibull S-N Fatigue Strength Curve Analysis for A572 Gr. 50 Steel, Based on the True Stress—True Strain Approach"

_applsci, doi:10.3390/app10165725_

Round 1

Reviewer 1 Report

Dear Author,

You find the reviewer's notes in the attached PDF file; please consider them.

Author Response

We are very grateful for the review.

We will hope feedback.

Sincerely,

Alejandro Molina

Reviewer 2 Report

The paper “Weibull S-N Fatigue Strength Analysis for ASTM A572 Steel, based on the True Stress-True Strain Approach” is devoted to the study of the fatigue strength characteristics of structural steels using ASTM A572 as an example. The article is certainly relevant, since, as the authors mentioned, there is a need to create a database of probability S-N curves for materials, which will be useful for structures design.

There are two remarks to the approach on the whole and to this article in particular:

  1. It should be noted that it is impossible to create a single probability S-N curve for the fatigue behavior of a material. The material of the same grade can vary in a certain range in terms of chemical composition, including by the content of harmful impurities (sulfur and phosphorus for ASTM A572) and microstructure parameters (for example, grain size, volume fraction of non-metallic inclusions, etc.). The influence of these parameters on the short-term strength characteristics and on the fatigue characteristics is different. For example, the increased sulfur content has little effect on the characteristics of short-term strength used in the authors’ calculations (yield strength, ultimate tensile stress), but it significantly reduces the resistance of the metal to fatigue. In this regard, it would be interesting to give several probability curves of fatigue failure for the same material, but having different parameters of the microstructure and chemical composition within the limits allowed by the standard. Of course, such studies are very time-consuming, but only after them it will be possible to talk about the real correct application of such curves in the design of structures.
  2. Also, as follows from the article (line 228), the authors used experimental data, obtained earlier by Arasaratman et. al. Of course, it is better to publish own experimental data, although taking into account the high complexity of fatigue testing, this is acceptable.

The aforesaid comments are in the nature of wishes, therefore, I recommend the article for publication in its current form.

Author Response

The paper “Weibull S-N Fatigue Strength Analysis for ASTM A572 Steel, based on the True Stress-True Strain Approach” is devoted to the study of the fatigue strength characteristics of structural steels using ASTM A572 as an example. The article is certainly relevant, since, as the authors mentioned, there is a need to create a database of probability S-N curves for materials, which will be useful for structures design.

There are two remarks to the approach on the whole and to this article in particular:

1. It should be noted that it is impossible to create a single probability S-N curve for the fatigue behavior of a material. The material of the same grade can vary in a certain range in terms of chemical composition, including by the content of harmful impurities (sulfur and phosphorus for ASTM A572) and microstructure parameters (for example, grain size, volume fraction of non-metallic inclusions, etc.). The influence of these parameters on the short-term strength characteristics and on the fatigue characteristics is different. For example, the increased sulfur content has little effect on the characteristics of short-term strength used in the authors’ calculations (yield strength, ultimate tensile stress), but it significantly reduces the resistance of the metal to fatigue. In this regard, it would be interesting to give several probability curves of fatigue failure for the same material, but having different parameters of the microstructure and chemical composition within the limits allowed by the standard. Of course, such studies are very time-consuming, but only after them it will be possible to talk about the real correct application of such curves in the design of structures.

1. Dear reviewer,

We are completely agree with you. However, although more research must be undertaken on this topic, we hope the proposed methodology can be used as a guideline to develop new theory. This because the incorporation of the mentioned differences on the material´s parameters of the micro structure and chemical composition can be made by complementing the first seven steps of the proposed methodology where the objective is to accurately determine the minimum and maximum expected strength values. Thus, the remains steps (8-15) will be minimally affected and consequently they can be used as guidelines to determine the corresponding S-N curves. But more research must be undertaken.

2. Also, as follows from the article (line 228), the authors used experimental data, obtained earlier by Arasaratman et. al. Of course, it is better to publish own experimental data, although taking into account the high complexity of fatigue testing, this is acceptable.

The aforesaid comments are in the nature of wishes, therefore, I recommend the article for publication in its current form.

2. We are completely agree with you, and moreover, due to our lab and time restriction as well as due to the objectives of the doctoral thesis of incorporating the Weibull analysis to the actual methodology, we decide to use the excellent data given by the Arasaratnam et. al. (2011).

Thanks by your invaluable comments.

Sincerely,

Alejandro Molina

Reviewer 3 Report

The paper is devoted to the method of measuring the fatigue strength of steel. In General, the research topic is interesting and may be of interest to a certain circle of readers. However, before the article is accepted, it must be thoroughly reworked.
1. The current version of the article resembles a scientific report, not a research paper. The article cannot be published in this form; it is necessary to take the presentation of the material more seriously.
2. It is necessary to adhere to the structure of research work. The chapters "Introduction", "Methods and materials", "Results and discussion", "Conclusion" should be present in the article. The conclusion should describe the results point by point. The current version of the conclusion is written blurry, it should be structured.
3. There are certain questions about the figures. For example, figure 3. What does it represent? My opinion is that the figures should be improved and described in more detail.
General conclusion: The article should be thoroughly revised and re-submitted for review.

Author Response

The paper is devoted to the method of measuring the fatigue strength of steel. In General, the research topic is interesting and may be of interest to a certain circle of readers. However, before the article is accepted, it must be thoroughly reworked.

1. The current version of the article resembles a scientific report, not a research paper. The article cannot be published in this form; it is necessary to take the presentation of the material more seriously.

1. Dear reviewer,

After reading our paper carefully, we conclude that because the main contribution of the paper is the formulation to use the Weibull analysis to determine the S-N percentiles (steps 8-15), the designed reliability (section 5.1) and the use of the derived S-N percentiles to perform the failure theory. Thus, because the actual structure let readers reach sequentially each phase then we believe the actual structure is correct. However, in the conclusion section the second point mention this structure. In the proposed methodology, steps 1-7 let us to determine the maximum and minimum strength S-N values based on which the Weibull parameters are determined. Steps 8-12 let us to determine the b and η parameters as well as the log-standard deviation used to determine the S-N percentiles. Steps 13-15 let us to determine the desired S-N percentiles.

2. It is necessary to adhere to the structure of research work. The chapters "Introduction", "Methods and materials", "Results and discussion", "Conclusion" should be present in the article. The conclusion should describe the results point by point. The current version of the conclusion is written blurry, it should be structured.

2. Since, experimental data was adopted from Arasaratnam et.al. (2011), then the main contribution of the paper is the formulation to use the Weibull distribution directly from the maximum and minimum addressed strength data (steps 8-15), thus we consider the actual paper structure is correct (others reviewer do not mention the paper´s structure). However, we agree with you our conclusions are blurry. Thus, the conclusion section was restructured to present the key findings point by point. Thank you by your comments.

3. There are certain questions about the figures. For example, figure 3. What does it represent? My opinion is that the figures should be improved and described in more detail.

3. Similarly, as a result of the reviews it was changed to an expanded and clearly figure to show the amplitude of the probabilistic S-N curve. It is important to highlight that, the figure represents the confidence level of reliability of the application case which is estimated on the fatigue strength cycle Sf103 based on the constructed S-N of the figure 2.

Thanks by your invaluable comments.

Sincerely,

Alejandro Molina

Round 2

Reviewer 1 Report

Dear Author, thank you for the performed corrections.

Author Response

Dear Author, thank you for the performed corrections.

Dear reviewer,

Thanks for your invaluable comments.

Sincerely,

Alejandro Molina

Reviewer 3 Report

Dear authors,
I do not doubt the significance of your work, but I would like to note that the comments were not corrected. The calculations and description in the manuscript are difficult to follow. In my opinion, the number of equations and calculations can be reduced. The manuscript should be presented as a summary of the work performed. Requirements for the manuscript are presented in the section "Instructions for authors" on the MDPI website. Please familiarize yourself with them.

Author Response

Dear authors,

Com1. I do not doubt the significance of your work, but I would like to note that the comments were not corrected.

Dear reviewer, your comments were

  1. Paper’s structure. After read our paper carefully we commitment its actual form is correct, thus it remains. Also by searching the journal content, we find recent papers with similar structure.
  2. To use the standard format of a research paper. Since we do not perform any experimentation then the usual structure methods and materials do not apply. And since the given formulation holds, then we do not consider necessary to add a discussion section.
  3. Restructure the conclusion section. Following your comment, it was restructured to present the main finding point by point.
  4. Since the objective of Figure 3 is showing readers the scheme of the derived S-N percentiles, then it was improved. The improvements can be seen in this submission round.

In general we believe the actual structure let readers to understand the paper content.

Com2. The calculations and description in the manuscript are difficult to follow.

We check the paper carefully, correct several minor grammar mistakes, and rework Figures and so on. And sections 4 and 5 were sequentially structured to present step-by-step the formulation and the application. Finally in the conclusion section readers can find the summary of the main points.

Com3. In my opinion, the number of equations and calculations can be reduced.

Dear reviewer,

Since the given formulation, (equations) and calculations let not expert readers to use them as guidelines to understand or to use them to search for deeper knowledge on the Weibull/Gumbel and non-homogeneous poison process, or to applied the proposed method to their own projects, then they remain.

Com4. The manuscript should be presented as a summary of the work performed. Requirements for the manuscript are presented in the section "Instructions for authors" on the MDPI website.

Dear reviewer,

We are agreeing with you, the presented article is a summary of a long development of the doctoral thesis that is been making. As well as, after read the “Information & guidelines for authors”, we consider that this article applies.

Please familiarize yourself with them.

Thanks for your invaluable comments.

Sincerely,

Alejandro Molina